# The Role of Microsphere Structures in Bottom-Up Bone Tissue Engineering

**DOI:** 10.3390/pharmaceutics15020321

**Published:** 2023-01-18

**Authors:** Ziyi Feng, Xin Su, Ting Wang, Xiaoting Sun, Huazhe Yang, Shu Guo

**Affiliations:** 1Department of Plastic Surgery, The First Hospital of China Medical University, No. 155, Nanjing North Street, Heping District, Shenyang 110002, China; zyfeng@cmu.edu.cn (Z.F.); 2020120820@cmu.edu.cn (X.S.); twang@cmu.edu.cn (T.W.); 2School of Forensic Medicine, China Medical University, No. 77, Puhe Road, Shenyang 110122, China; 3School of Intelligent Medicine, China Medical University, No. 77, Puhe Road, Shenyang 110122, China; hzyang@cmu.edu.cn

**Keywords:** microspheres, bone tissue engineering, organoid, endochondral ossification, anti-inflammatory

## Abstract

Bone defects have caused immense healthcare concerns and economic burdens throughout the world. Traditional autologous allogeneic bone grafts have many drawbacks, so the emergence of bone tissue engineering brings new hope. Bone tissue engineering is an interdisciplinary biomedical engineering method that involves scaffold materials, seed cells, and “growth factors”. However, the traditional construction approach is not flexible and is unable to adapt to the specific shape of the defect, causing the cells inside the bone to be unable to receive adequate nourishment. Therefore, a simple but effective solution using the “bottom-up” method is proposed. Microspheres are structures with diameters ranging from 1 to 1000 µm that can be used as supports for cell growth, either in the form of a scaffold or in the form of a drug delivery system. Herein, we address a variety of strategies for the production of microspheres, the classification of raw materials, and drug loading, as well as analyze new strategies for the use of microspheres in bone tissue engineering. We also consider new perspectives and possible directions for future development.

## 1. Introduction

Bone defects cause immense economic and healthcare concerns throughout the world. Reparation of large bone defects, as well as the restoration of the anatomical structure, appearance, and function of the injured area, has presented major difficulties [1]. At present, autologous and allogeneic bone transplantation remain the “gold standard” of bone substitution therapy in clinics. However, the drawbacks are numerous and well-known. Patients of autologous bone transplantation must undergo additional surgery, and there is a limited amount of autologous material, which in some cases may be insufficient for large-area bone defects. Additionally, there is a high incidence rate of complications such as fractures, nonunion, and infection after allogeneic bone transplantation. Therefore, it is necessary to devise new methods for engineering tissues that mimic native bone [2].

Bone tissue engineering is an interdisciplinary biomedical engineering method that can provide solutions to these complications, as it can act as an alternative to tissue transplantation [3]. This method involves the separation, cultivation, and expansion of autologous high-concentration cells in vitro, then their implantation on natural or synthetic cell scaffolds, or into the bone defect site [4]. With the gradual degradation of biomaterials, the implanted cells proliferate and differentiate, forming new bone tissue. This, in turn, helps to retain, maintain, and improve the functionality of damaged sites. Engineered bone tissues have the possibility to not only exert these functions at the time of treatment but also to continue the formation of functional tissue as needed after implantation and to further mature in situ [5]. The three elements of bone tissue engineering are “scaffold materials”, “seed cells”, and “growth factors”, the last of which can promote the growth and differentiation of seed cells [6,7].

Cells applied in the reconstruction of tissues and organs are collectively referred to as seed cells. In recent years, MSCs have been widely studied for use as seed cells in bone tissue engineering. MSCs, which were discovered by Friendenstein et al. in 1968, are widely distributed in almost all tissues of both fetuses and adults, including bone marrow, blood, the umbilical cord, umbilical cord blood, placenta, fat, the amniotic membrane, amniotic fluid, dental pulp, skin, and menstrual blood [8,9]. Bone marrow and adipose tissue are the most common cell sources of MSCs. MSCs not only have the potential of multi-directional differentiation, but in recent years, studies have shown that MSCs can promote vascular regeneration [10], reduce apoptosis [11], and inhibit the occurrence of inflammatory response [12].

The foundation for successful tissue construction is the interaction between the microenvironment and the seed cells [13]. However, the monolithic structure of the traditional scaffold limits mass absorption of nutrients and gas molecules such as ions, growth factors, and oxygen. Furthermore, the removal of cell metabolite by-products is a problematic issue, as well as the uneven distribution of cells and the extracellular matrix and necrosis within the structure [14]. Moreover, this “top-to-bottom” strategy is not adequate to flexibly adapt to variations in the shapes of some defects [15,16]. With the development of micro/nano biological manufacturing technology and biomaterials, a new kind of “bottom-up” modular tissue engineering technology was proposed as a way to construct three-dimensional functional tissues by combining micro repetitive functional units to curb those issues [17,18,19]. A simple but effective solution using this “bottom-up” method was put forward in the 1990s that involves introducing microspheres into a continuous matrix instead of compromising the properties of the bulk scaffold [20]. Microspheres are structures with diameters ranging from 1 to 1000 µm, and they can be categorized according to structural differences as solid microspheres, porous microspheres, or hollow microspheres. The solid and the porous types are most commonly used in seed cell culture processes, where the particulates act as bioreactors and exert therapeutic effects by modulating cell behavior [21]. Furthermore, their inherently small size, high drug loading efficiency, infusibility, and high reactivity to the microenvironment enable the hollow microspheres to be effective as a controlled drug delivery tool [22,23].

Microspheres have a number of unique properties compared to large pieces of material materials that make them attractive for biomedical applications. Microspheres can be fabricated from both natural and synthetic materials, and can be made to vary in shape, density, porosity, and size by applying techniques that are often compatible with the encapsulation of biologics. Microspheres for different application scenarios require different properties [24]. Microspheres can be divided into three categories: suspensions, granular and composites. In suspensions, the microspheres reside in a fluid (liquid or air), with minimal interactions between particles. When the particle-packing density increases, granular microspheres form. If microspheres are embedded within a bulky material, a composite is obtained [25]. Physical interactions between microspheres typically result in shear thinning behavior and solid consistency without chemical modification after injection. In addition, microspheres are inherently modular, because multiple microspheres populations can be mixed together to create different materials with different properties [26].

This review addresses a variety of strategies for the fabrication of microspheres that allow for well-controlled morphology, particle size, composition, and surface modification. Following this, we introduce the microspheres formed by biomaterials into delivering cells and the construction of micro modular bone tissue, mainly focusing on the role of microspheres as drug carriers, bioreactors, and reservoirs. Through the comprehensive review of previous studies, we examine the current challenges faced by bioengineers using bottom-up methods of bone tissue engineering, as well as perspectives on possible directions for future development.

## 2. Fabrication of Microspheres

### 2.1. Emulsion Polymerization

Emulsion polymerization is a classic method for preparing microspheres. By utilizing an emulsifier and mechanical agitation, the water-soluble monomers are dispersed in water, thus forming an emulsion. Initiate polymerization is a special method of free radical polymerization [27]. The reaction system consists of a hydrophobic monomer, a water-soluble initiator, an emulsifier, and water. With water as the dispersion medium, the free radical generated by the initiator enters the monomer swelling micelle from the aqueous phase and reacts with the monomer to form a nucleus. The hydrophobic monomer then diffuses from the droplet through the aqueous phase. In the process, the free radical polymerizes with the nucleus generated by the monomer. With the steady growth of the nucleus, it eventually becomes a microsphere and precipitates until the monomer diffusion ends and the droplet disappears. However, inside the microsphere, the monomer continues to polymerize until the reaction ends. Microspheres prepared using this method have high homogeneity and monodispersity, alongside high polymerization speed and molecular weight [28]. The spherical particles are formed mainly because of the interfacial tension; however, in one study, Fab et al. used negatively charged hydrophilic acrylic acid as a comonomer and attempted to introduce the hydrophilic and hydrophobic vinyl monomer into the oil-water interface system. The emulsion interface of the water (hydrophilic monomer aqueous solution) and oil (containing the hydrophobic monomer) was subsequently constructed, and the anisotropic Janus microspheres with hydrophile lipophilic properties were successfully formed by emulsion polymerization. This method can not only synthesize anisotropic Janus microspheres, but also synthesize spherical microspheres with porous structure, offering an entirely new direction for emulsion polymerization [29].

Other types of emulsion polymerization include core shell emulsion polymerization, soap-free emulsion copolymerization, and microemulsion polymerization. Physical methods such as an ultrasonic magnetic field have been proposed, but they are still in the theoretical stage [30].

### 2.2. Suspension Polymerization

Suspension polymerization refers to the method of free radical polymerization in which the monomer is dissolved with the initiator and suspended in water in the form of droplets, with the water as the continuous phase and monomer as the dispersed phase. Suspension polymerization has a high polymerization rate and requires a high relative molecular mass of the microspheres, whereas the impurity content is far lower than that of emulsion polymerization [31]. Using this method, researchers have successfully synthesized nano monodisperse crosslinked polystyrene microspheres without any chemical derivation or grafting steps. The nano microspheres synthesized using this method have a good shape and no impurities on the surface [32]. In addition to smooth microspheres, porous polymer microspheres have been widely used in the immobilization of enzymes and catalysts, adsorption and separation, detection, and sensing due to their high specific surface area, abundant pores, and good stability. In one case, Zhang et al. successfully prepared surface folded porous polymer microspheres in the traditional suspension polymerization system with the help of phase separation and volume shrinkage [33].

### 2.3. Precipitation Polymerization

The microspheres obtained via precipitation polymerization not only have narrow particle size distribution and a simple operation process, but they are also easily obtained for follow-up treatments. Unlike other preparation methods, the precipitation polymerization process does not use an additional dispersion stabilizer, so the reaction conditions must be accurately controlled. Additionally, because of the absence of the dispersion stabilizer, and thus the absence of the need to separate it, it is safer for application in the biomedical system [34]. In their study, Yin et al. prepared functional gel microspheres with high adsorption capacity for cationic toxins through the in situ crosslinking polymerization of precipitate droplets. This preparation method is quite simple to achieve. First, a homogeneous aqueous phase reaction solution (monomer, initiator, crosslinking agent, and water) is prepared and then dropped into hot corn oil that has been heated by externally circulating water at 90 °C. Using this method, polymerization is completed in a very short time [35].

One noteworthy type of precipitation polymerization is called dispersion polymerization. Because of its unique polymerization mechanics, dispersion polymerization is widely used in the preparation of various types of polymer microspheres with a size of 0.1–15 μm in diameter [36]. Unlike other methods of precipitation polymerization, stabilizer is needed in the dispersion polymerization system. Precipitation polymerization is the formation of polymers that are insoluble in the mutual poly-sink system [37]. When the length of the oligomer chain generated by the reaction reaches a critical value, the polymer cannot remain dissolved in the reaction medium, allowing polymer microspheres to be obtained. The microspheres are characterized by having a good spherical shape, large particle size (compared with those of emulsion polymerization), narrow particle size distribution, and low viscosity. Precipitation polymerization is commonly used in the preparation of functional microspheres. The dispersion copolymerization reaction system, on the other hand, is very complex, and different comonomers form different polymerization systems, resulting in different reaction behaviors [34].

### 2.4. Spray Drying Method

Spray drying methods can be categorized as the pressure spray drying method, centrifugal spray drying method, or airflow spray drying method. Sprinklers, desiccators, preheaters, air separation chambers, air filters, collecting bins, and blowers are common components of spray drying. The goal of this method is to disperse the given materials as a mist of tiny droplets, expose them to a hot air flow, and evaporate most of their water, thus obtaining a powder, fine-granular finished product, or semi-finished product [38]. Quinlan et al. encapsulated vascular endothelial growth factor (VEGF) in alginate microparticles by spray drying, producing particles less than 10 µm in diameter. Using this process, they achieved effectively encapsulated and controlled VEGF release for 35 days, which was sufficient to increase tubule formation by endothelial cells in vitro [39].

### 2.5. Microfluidics

Microfluidics is a technology that uses microchannels to manipulate micro fluids. This method integrates one or more functions of a large laboratory on a micro or even nanoscale chip. By effectively and intelligently encapsulating cells or drugs in microspheres, the successful management of their functions and characteristics can be achieved. This process can also result in the development of some unique characteristics, which promote multi-disciplinary exchanges and cooperation, as well as the development of precision medicine, new manufacturing technology, and applied materials. In recent years, the use of microfluidic technology has shown many advantages in the preparation of microspheres. Compared with conventional preparation methods, the microfluidic preparation platform can more accurately control reaction conditions such as temperature and pressure with fast preparation speed and good reproducibility, while also using less raw materials and reagents [40]. More importantly, the size, morphology, and dispersion of particles can be accurately regulated using the microfluidics management preparation platform [41]. The main preparation methods of microspheres by microfluidics at present are the droplet template method and the flow lithography method. The droplet template method refers to the preparation of microdroplets through the introduction of fluid shear, electric field induction, centrifugal throwing, formation of microspheres in the system as a template, and proper curing. Flow lithography is a micro projection technology based on photopolymerization in multiphase laminar flow. It involves irradiating the fluid in the microfluidic chip with an ultraviolet beam in a preset shape, causing the irradiated fluid to partially polymerize according to the beam shape, thus obtaining homogeneous microspheres. The whole preparation process can be completed with the help of a commercial inverted fluorescence microscope [42].

Other studies have used sodium alginate to encapsulate single mesenchymal stem cells (MSCs) in a microfluidic platform, thus simulating a three-dimensional microenvironment to support cell viability and function, while at the same time protecting cells from environmental stresses. Compared with acellular microgels, the MSCs’ loaded alginate gel microspheres showed significant enhancement in bone formation in a rat tibia ablation model, which lays the foundation for modular bone tissue engineering [43]. In addition to the encapsulation of cells, microfluidics-controlled hydrogel microspheres provide a good supporting environment for cell adhesion and growth. In their study, Cui et al. constructed a bisphosphonate functionalized injectable hydrogel microsphere (GelMA-BP-Mg) using a metal ion ligand coordination reaction to achieve Mg^2+^ loading and controlled release via magnetics, thereby promoting cancellous bone remodeling in osteoporotic bone defects. The results of the in vivo and in vitro experiments show that the composite microspheres inhibit osteoclasts by stimulating osteoblasts and endothelial cells. This is conducive to osteogenesis and angiogenesis, which effectively promote cancellous bone regeneration [44]. Meanwhile, an efficient transfer platform was designed using new one-step microfluidic technology to achieve the combination of liposomes and photo crosslinked gel matrixes, thus forming monodisperse gel/liposome mixed microgel rapidly under ultraviolet light. Microgel can release kartogenin (KGN) continuously during the degradation period, thus providing a feasible method for the treatment of osteoarthritis in the joint release of KGN [45]. In this review, we aim to provide a discussion on the reported particle design methods that can modulate their (bio)chemical/physical and structural characteristics (Figure 1).

## 3. Microsphere Manufacturing Materials

Primarily biocompatible and biodegradable materials are used for the preparation of sustained-release microspheres. Scientists use natural and synthetic polymers to achieve the purpose of drug loading and controlled release. Of course, in order to be suitable for different preparation methods, the characteristics of selected materials must also vary. In order to achieve different trial purposes, it is usually necessary to add additional materials such as therapeutic drugs, fluorescent tracers, photosensitizers, etc. These additional excipients also have biocompatibility and biodegradability requirements.

### 3.1. Natural Materials

The natural biomaterials used are mainly chitosan, collagen, starch, liposomes, and lipoproteins [46].

Chitosan has unique adhesion properties and can encapsulate drugs. After administration, it degrades quickly in the body, so its toxicity and side effects are low, and it can be metabolized by specific enzymes in the human body. Chitosan has malleable properties and can thus be modified into various shapes and sizes, and various chitosan-based biomimetic organic–inorganic composite materials are available in different forms (hydrogels, fibers, porous scaffolds, microspheres, etc.) that can be applied to bone tissue engineering and enamel or dentin bionic restoration [47]. Chitosan-based materials are biocompatible and have the potential to release BMP-2 systematically and sustainably when needed. This release results in increased levels of cell proliferation, enhanced alkaline phosphatase activity, increased differentiation, and increased mineralization under in vitro and in vivo conditions [48]. Chitosan can have its physical properties modified by various chemical modifications, such as carboxymethylation, thiolation, and succinylation. Currently, synthesized chitosan derivatives have improved solubility, increased cellular uptake, reduced cytotoxicity, encapsulated different types of drugs, and sustained release properties [49]. In addition, chitosan of different molecular weights has different biological efficacy. For example, a study showed that microspheres of high molecular weight chitosan exhibited lower adhesion and lower MTX release compared to medium molecular weight chitosan rates [50].

Another natural polymer material commonly used in the preparation of microspheres is gelatin. It has good film-forming properties, its properties change according to temperature, and it has a gelatinous quality. It is also suitable for facilitating the release of drugs and growth factors [51,52]. Scientists have found that the drug-retaining, composite system of collagen promotes bone regeneration more effectively than empty microspheres [53]. A study by Dennis P Link et al. showed that gelatin type affects the degradation of gelatin particles incorporated into calcium phosphate (CaP) cement. However, this difference in degradation and the consequent macroporosity did not induce a difference in biological response [54]. Gelatin and chondroitin sulfate microspheres can be degraded by several gelatinases found in the synovial fluid of osteoarthritic joints [55]. In addition, enzymatic degradation of gelatin/HA microspheres may lead to reduced calcium uptake by reducing the integrity of the composite microspheres [56].

Alginate-based materials have received considerable attention in biomedical applications due to their hydrophilicity, biocompatibility, and physical structure. Applications include cell encapsulation, drug delivery, stem cell culture, and tissue engineering scaffolds [57]. Alginates have been analyzed for tissue engineering applications along with bone morphogenetic proteins, vascular endothelial growth factor, transforming growth factor β-3, other growth factors, cells, proteins, drugs, and osteoinductive agents [58]. For example, several authors have developed injectable hybrid RGD-alginate/Laponite hydrogel microspheres to co-encapsulate human dental pulp stem cells and VEGF. The microspheres exhibited tunability in terms of mechanical properties and sustained-release capability capacity. The incorporation of hectorite and sustained release of VEGF supported not only the differentiation of dental pulp stem cells in vitro but also the regeneration of new tissues in vivo [59].

Chitosan, collagen, and starch are the matrix components of the tissue, all of which have good biocompatibility and biodegradability, low cytotoxicity, and degradation products that can be completely absorbed by the body. However, the disadvantage with these components is that they have poor mechanical strength, and their degradation rates are not consistent. Sophie R. Van Tommeden et al. reported on the degradation behavior of in situ gelling hydrogel matrices consisting of positively and negatively charged dextran microspheres. They found that the degradation of positively charged microspheres was faster compared to negatively charged microspheres due to the presence of protonated tertiary amine groups in the cationic microspheres, which stabilize the transition state during hydrolysis. On the other hand, the presence of negatively charged groups leads to the repulsion of hydroxyl anions, which results in a slower degradation rate [60].

### 3.2. Synthetic Materials

With the development of materials, science, and technology for preparing microspheres, synthetic materials are increasingly sought by clinicians and scientists.

The most commonly used synthetic materials are polyester materials that have been approved by the US FDA and are safe for medicinal use. These include polylactic acid (PLA), polyglycolic acid (PGA), polyvinyl alcohol (PVA), polylactic acid glycolic acid copolymer (PLGA), and polycaprolactone (PCL).

Among them, PLA and PLGA are widely used in sustained and controlled release injection drug delivery systems due to their good biocompatibility and biodegradability. Usually, when they are exposed to an aqueous environment, the ester bond is hydrolyzed and the molecular weight decreases. The degree of cross-linking thus decreases, and when the molecular weight decreases to the point of being soluble in water, the material is gradually destroyed to form metabolizable lactic acid and α-glycolic acid. PLA and PLGA are widely used in osteogenic regeneration [61,62,63].

Other synthetic microsphere materials such as poly(ethylene glycol) (PEG) or its derivatives and the combinations of multiple polymers are also favored by scientists. For example, some scientists have used a combination of chitosan and polyethylene glycol diacrylate (PEGDA) to use chondrocyte-loaded microspheres as a cell carrier based on a double network hydrogel. Applying microfluidic technology, size-controlled chitosan/PEGDA hydrogel microspheres were fabricated by the water-in-oil method after photo-crosslinking and physical crosslinking. Chondrocytes loaded on microspheres showed good cell viability and proliferation after long-term cell culture [64]. Microspheres with surface micropores were obtained by using an amphiphilic triblock copolymer (PLLA-PEG-PLLA) consisting of poly(L-lactic acid) (PLLA) and PEG segments. The W1/O/W2 double emulsification method was used. When the PEG fraction was controlled at 10 wt.%, the microspheres exhibited higher cell affinity than the smooth-surfaced PLLA microspheres. After further functionalization with polydopamine coating and apatite deposition, PLLA-PEG-PLLA microspheres significantly enhanced the osteogenic differentiation of bone marrow MSCs [65].

### 3.3. Composite Materials

In order to achieve loading of multiple drugs and other complex requirements, researchers usually use composite materials to achieve goals that cannot be achieved using single materials. For example, a variety of materials are composited to make microspheres. To improve the performance of microspheres [66,67], various materials are combined, thus adjusting the microenvironment [68,69]. The production of multi-layer microspheres also requires the compounding of multiple materials in order to achieve the purpose of carrying and releasing multiple drugs [70].

### 3.4. Auxiliary Additives

For varying purposes, in addition to loading the drugs, scientists often add some auxiliary materials to change the properties and functions of the microspheres by surface modification or incorporation. These may include: fluorescent agents or radionuclides to trace [71], photosensitizers to improve photosensitivity properties [72], HA to increase osteogenic efficiency and the strength of materials [73], incorporation of Nintedanib into folate modified albumin microspheres to improve water solubility and the targeting effect of the drug [74], GRGDSPC to increase the hydrophilicity and cellular affinity of microspheres [75], and others.

## 4. Application of Microspheres in Cargo Delivery

In bone tissue engineering, microspheres are used as carriers of growth factors and drugs, and they can be effectively used in delivery. As they are small and spherical, they can be injected directly into the treatment site [76]. Therapists can adjust the drug release rate by modifying the size and composition of the microspheres to achieve steady long-term drug release. For example, with an increase in the diameter of monodisperse PLGA microspheres, there is a decrease in the drug release rate and a prolongation of its action time [77]. Compared with oral treatment, injected microspheres can release a drug at a constant rate, have fewer side effects, and can reduce the systemic toxicity of the drug [78]. Moreover, microspheres can better protect the encapsulated drug, thereby avoiding the degradation of enzymes in the receptor caused by gastric and intestinal juices [79]. Researchers can also design microspheres with different materials according to the specific in vivo environments, such as p(BMA-co-DAMA-co-MMA), lipids, and other phase changing materials [80]. However, it is difficult to reproduce and predict the full impact of the human internal environment when designing the microspheres, and sometimes it is difficult to maintain a constant release rate in the body. Despite these limitations, microspheres as a drug or growth factor carriers have been found to be effective in the treatment of bone defects [81,82]. This holds true for all the various drugs that have been used with this method, which include hormones, growth factors, antibiotics, anti-tumor drugs, biologics, metal ions, and others. All of these will be discussed in depth in this section (Figure 2), (Table 1).

### 4.1. Hormones

Since the 1950s, corticosteroids, a type of steroid hormones, have been regularly used to treat symptoms associated with arthritis [128]. At the same time, corticosteroids such as dexamethasone have also been used as an inducer of osteogenic differentiation, which can promote bone regeneration in inflammatory conditions [129]. Different doses of corticosteroids have different effects on the regulation of the inflammatory microenvironment and the induction of cell osteogenic differentiation [130]. Corticosteroids such as triamcinolone acetonide (TAA) exhibit maximum efficiency when delivered directly to the site of inflammation. Due to their short half-life in vivo, microspheres can be applied to prolong their therapeutic effect [131].

Melatonin was first isolated by Aaron B. Lerner and his colleagues in 1958. It is a biological agent that comes mainly from the pineal gland, but it can also be synthesized and secreted by tissues and organs such as the retina, skin, liver, intestine, ovary, testis, and bone marrow. Under the influence of the circadian rhythm, melatonin has various physiological functions, including anti-aging, antioxidant, analgesic, and hypnotic effects [132]. There are two processes through which melatonin regulates inflammation: the first is through the inhibition of oxidative stress, which causes the regulation of a wide range of molecular signaling pathways and can reduce the occurrence and development of inflammation under various pathophysiological conditions. The second method is receptor independent, and works by the melatonin binding to transition metals, thus inhibiting the formation of hydroxyl radicals and regulating the transcription and activity of antioxidant enzymes [133,134]. Moreover, studies have confirmed that melatonin can synergistically play an anti-inflammatory role in a variety of diseases by acting on MSCs and their derived exosomes [135]. In recent years, melatonin has also been widely considered a potential bioactive molecule for promoting stem cell osteogenesis. This is because it has been proven to regulate bone formation and resorption, as it can promote the differentiation of osteoblast precursors into osteoblasts by affecting the Wnt signaling pathway. At the same time, it can inhibit the differentiation of osteoclasts by increasing the expression of osteoprotegerin (OPG). This antagonizes the RANKL induced NF-κB pathway, thus significantly promoting osteogenesis in the melatonin-supplemented osteogenic medium [136]. For example, one study found that the reduced secretion of melatonin in elderly mice may be an inducing factor in osteoporosis [137]. In another study, Gurler et al. wrapped melatonin with PCL particles. After being suddenly released initially, the melatonin was able to maintain sustained release within 8 h. By comparison, the authors confirmed that the particles of 3 wt% PCL solution had the most suitable particle size for melatonin encapsulation (2.3 ± 0.64 μm). Other in vivo and in vitro experiments have confirmed that melatonin loaded PCL microspheres are capable of promoting bone regeneration [109]. PLGA and chitosan are also used in the encapsulation and application of melatonin, respectively, and have been proven to promote bone formation [138,139].

### 4.2. Growth Factors

Growth factors are natural proteins that can be used to stimulate cell growth. Some of them can be used for bone regeneration in tissue engineering, as they regulate bone development and the osteogenic differentiation of stem cells. In line with our research, microspheres are an efficient delivery method for growth factors. Growth factors involved in bone formation can be divided into two categories: the first one is the serine threonine-kinase receptor, which has a high affinity with TGF-βs, as well as BMPs growth factors. The second type is tyrosine kinase receptors, which play a specific role in the growth factors FGF, VEGF, PDGF, and IGF [140].

The half-life of BMP2 in a physiological environment is very short, lasting only 7 min. Due to its instability and rapid degradation rate, it is necessary to load high doses into scaffolds. However, this increases costs and side effects such as inflammatory response, nerve injury, ectopic ossification, and the development of tumors. In order to avoid these issues and achieve more effective bone regeneration, Chen et al. modified BMP2 and wrapped it with gelatin microspheres [141]. In another study, Kang et al. designed a composite consisting of gelatin microparticles loaded with VEGF/FGF-2 and poly(lactic-co-glycolic acid)-poly(ethylene glycol)-carboxyl (PLGA-PEG-COOH) microparticles loaded with BMP-2 encapsulated in a nano hydroxyapatite-PLGA acid scaffold. Using this method, they were able to achieve the combined effect of osteogenesis and angiogenesis [142]. TGF-β1 is a secretory protein involved in cell growth, cell proliferation, and cell differentiation.

Whereas injectable calcium phosphate/gelatin particles on their own showed excellent osteogenic properties and mechanical strength, the addition of TGF- β1 significantly enhanced the bone reconstruction process [143]. PEX and PLGA can also be mixed, causing particles to achieve controlled release and enhancing bone marrow stromal cell proliferation and osteoblast differentiation in vitro [144].

Large bone regeneration is a challenge in surgery. A key step in the process of bone formation, endochondral osteogenesis (EO) has become a research hotspot in recent years [145]. TGF-β1 has also been proven to promote the formation of chondrocytes, either through exogenous delivery in the culture medium or release from the incorporated microspheres. These processes can induce the chondrogenesis of human mesenchymal stem cells [146]. Therefore, multiphase particles can be used to promote the continuous release of growth factors in order to achieve endochondral osteogenesis and develop new ideas and strategies for the repair of large-area bone defects [147].

### 4.3. Antibiotics and Anti-Tumor Drugs

Bone defects caused by inflammatory infection or tumor surgery are the source of many clinical problems. The anti-inflammatory and antibacterial requirements after stent implantation are also important issues that clinicians and researchers need to consider. For this reason, drugs encapsulated or carried by microspheres have become a useful option for physicians and scientists. Based on the specific situation, scientists have designed a variety of microsphere production methods to achieve an ideal drug carrying and controlled release process. The drug carrying capacity of microspheres has a unique effect that cannot be replaced by other methods of administration, such as oral administration and intravenous injection. First, the drug carried by the microsphere can directly contact the lesion, which reduces the first-pass clearance of the drug and the damage to other organs and tissues, making the drug more effective. Secondly, the drug carried by the microspheres can achieve controlled release and selective release in different stages of the disease, increasing its effect on the pathophysiological process and therefore achieving effects that cannot be achieved by other methods of administration. Finally, the microspheres carry the drug directly and expediently, which eliminates the need for repeated medical treatments, making the use of microspheres a more humane and customizable method.

Antimicrobial agents commonly used in bone tissue engineering are: gentamicin [148,149,150], vancomycin [151,152], ciprofloxacin [153], levofloxacin [154], tetracycline [155], metronidazole [156,157], doxycycline [157], antibiotic colistin [158], or antimicrobial peptide (Pac-525) [159]; or anti-tuberculosis drugs: isoniazid, rifampin [154], rifampicin [160]. Of course, they can also be used not just alone, but in combination with other drugs to reduce infection and promote osteogenic differentiation [149,150,152].

Bone cancer or metastatic cancer is a commonly occurring condition. In order to achieve the therapeutic objectives of improving treatment efficiency and overcoming drug resistance and toxicity, controlled release antitumor drugs are needed in bone tissue engineering. Some examples are the combination of methotrexate (MTX) with doxorubicin (DOX) [161], and the combination of paclitaxel (PTX) with etoposide (ETP) to treat osteosarcoma [105].

### 4.4. Gene Therapy and Exosomes

Gene therapy using viral vectors can be used to target specific molecules, thus enhancing the reparation of bone defects. The microspheres can deliver plasmid deoxyribonucleic acid (pDNA) to specific sites during the treatment of bone defects to increase the expression of important genes [162]. In a study conducted by Feng et al., pDNA and a hyperbranched polymer were self-assembled and encapsulated into PLGA microspheres. This allowed the researchers to transfect pDNA into target cells, control their release, and ensure high delivery efficiency. The microspheres in this study were co-injected with nanofiber sponge microspheres to locate the orphan nuclear receptor 4A1 and were thus able to play a therapeutic role [163]. In another study, McMillan et al. integrated TGF-β1 and BMP-2 pDNA composite in the liposomes, then embedded it in mineral-coated hydroxyapatite particles to successfully treat bone defects through the gene editing of MSCs [164].

MicroRNA (miRNA), an endogenous non-protein coding RNA with a length of about 22 nt, affects the output of many protein coding genes by specifically binding to the 3′UTR of the target gene. So far, about 2600 miRNAs have been identified as participatory in the regulation of more than 60% of all coding genes in the human body [165]. miRNAs usually bind themselves to the target mRNA in a partially paired way, which means that a single miRNA can regulate hundreds of target genes and have a great impact on many biological processes, including cell proliferation, migration, differentiation, or apoptosis. Researchers found that microRNAs can regulate multiple physiological functions by targeting multiple genes of the same or different signal pathways [166,167,168]. Though the continuous release of microRNA can play a vital role in creating an optimal osteogenic microenvironment, microRNA in vivo usually degrades quickly. Therefore, it is necessary to find a material that can achieve more effective long-term bone induction at a moderate degradation rate; microencapsulated microRNA hydrogel can be used as a primary therapy or adjuvant therapy to supplement surgical resection. In this vein, Gan et al. promoted the osteogenic differentiation of MSCs by encoding cholesterol modified non-coding microRNA and injectable PEG hydrogel [110].

In recent years, studies have shown that the paracrine effect of MSCs has broad therapeutic prospects. Siu et al. conducted a proteomic analysis on the substances secreted by bone marrow mesenchymal stem cells and found that the substances included a variety of membrane vesicles and cytoplasmic proteins. These regulate metabolism, defense response, and tissue differentiation in biological processes such as vascularization, hematopoiesis, and bone development. The results of this study show that human MSC-conditioned medium has the potential to treat a variety of human diseases, which lays the foundation for cell-free therapy [169,170]. As such, the exosomes secreted by stem cells have gradually become a research hotspot. It is believed that exosomes can produce therapeutic effects through a variety of mechanics. Exosomes are vesicles with a diameter of 30–150 nm, and they are rich in transcription factors, proteins, miRNAs, lncRNAs, and mRNAs. Contact between exosomes and receptor cells can occur through one of three mechanisms: (a) interaction with signal receptors on target cells; (b) release of contents after fusion with the plasma membrane of receptor cells; or (c) direct endocytosis by receptor cells, thus regulating a variety of biological functions through the regulation of signal pathways [171,172,173]. As exosomes are easy to remove in vivo, wrapping them in microspheres is a good strategy to improve their function.

For example, Swanson et al. prepared PLGA and PEG triblock copolymer microspheres wrapped with exosomes by using the microfluidic system to repair craniomaxillofacial bone defects, in which exosomes were located in the inner layer of PEG. At first, an explosive release occurred, followed by a nearly linear release curve which was able to provide a stable dose of exosome release for up to 10 weeks. Compared with the blank control and the non-encapsulated exosomes group, the exosomes encapsulated by microspheres showed better mineralization ability on the 21st day. Subsequent in vivo animal experiments further confirmed that microspheres including the exosomes had a positive effect on bone defect reparation [174]. In another study, Li et al. wrapped the exosomes of human urine derived stem cells with PLGA nanoparticles. These had a good therapeutic effect on osteolysis caused by wear debris, and the series of inflammatory factors secreted either macrophages or T lymphocytes after a total joint replacement [111].

### 4.5. Metal Ions and Others

Metal and metal ions are also used in bone tissue engineering. However, localized high concentration ion release often produces undesirable effects. Controlled release technology is needed to achieve the physiologically optimal concentration. The commonly used metals are Sr [175], Ag [176], Co ions [177], etc. Sr, for example, can be used in monomers or compounds for physiological effects, with the most commonly used compounds being strontium ranelate (SrR) and SrCl_2_. SrR is a type of strontium salt, which has a beneficial overall effect on the improvement of bone microstructure. However, high concentrations of SRR are often counterproductive to cell proliferation and osteoblastic differentiation, hence the need for controlled release options [175,178]. A similar effect has been obtained by coating SrCl_2_ [179].

Statins are often used as lipid-lowering drugs. However, several recent studies have found that they have several other non-lipid-regulating effects, such as the improvement of vascular endothelial function, an antioxidant effect, an anti-inflammatory effect, an easing of rejection reaction effect, anti-tumor effects, anti-Alzheimer’s disease effects, and many others, and they are also useful in the treatment of osteoporosis. They are therefore often used in bone tissue engineering, with the most commonly used types being simvastatin and lovastatin. Because of the side effects that can come with systemic administration, and because of the proangiogenic function of local high efficiency targeting, local controlled release can be a better option. Simvastatin may be used alone [180] or in combination with other drugs: dexamethasone [157,181], metronidazole [156], PDGF [182,183,184], ketoprofen [157], and doxycycline [157] to meet a variety of clinical needs.

In addition to the controlled release drugs detailed above, there are other more common drugs that are designed for use under appropriate clinical conditions. For example, salvianolic acid B (SB) promotes osteogenic differentiation of BMSCs [185]; KGN enhances the migration and chondrogenic differentiation of BMSCs [186]; oxygen carrier perfluorooctane (PFO), as a local oxygen source, increases cell viability and helps to maintain the osteogenic differentiation of human periosteum-derived cells (HPDC) under hypoxia conditions; kaempferol (KEM) improves bone formation [187]; and ketoprofen prevents tissue destruction [157].

## 5. Novel Applications of Microspheres for Bone Tissue Engineering

### 5.1. 3D Culture of Seed Cells and Construction of Organoid

To ensure the efficient proliferation and differentiation of seed cells in vivo, it is necessary to provide a cell scaffold that acts as an artificial extracellular matrix. The scaffold materials used in bone tissue engineering must have good biocompatibility and surface activity, bone conductivity and bone induction ability, and appropriate pore size and porosity, as well as moderate mechanical strength and plasticity [188]. At present, both biodegradable and non-biodegradable scaffold materials are widely used. However, due to the high levels of mechanical strength and strong biological inertia, non-biodegradable materials can cause secondary surgery problems. This has led to an increase in the study of biodegradable and bioactive materials in recent years [189].

As a microcarrier structure, microspheres have a larger surface area and volume ratio than monolayer cells, so they can effectively promote the growth of cells [190]. Microspheres are commonly used in 3D cell cultures in the process of stem cell growth and differentiation. This is because over the past decade, scientists have found that 3D cell culture technologies can provide cells with a culture environment more akin to the in vivo environment. Thus, the 3D cell culture must be able to simulate the main characteristics of the in vivo environment, including the interactions between cells and the extracellular matrix, between cells and organs, and between cells and other cells, in order to better realize the morphology and function of cells in vitro, understand their physiological function, and establish extensive cell-to-cell and cell-to-extracellular matrix (ECM) interactions [191]. At present, studies have confirmed that a 3D culture of stem cells with microspheres can efficiently promote the growth of stem cells as well as osteogenic and chondrogenic differentiation [192,193], and promote the paracrine ability of stem cells. This causes more anti-inflammatory factors and nutritional factors (such as VEGF, PDGF and FGF) to be expressed, thus allowing stem cells to play a more effective therapeutic role in bone regeneration [194]. Compared with traditional 2D cultures, the MSCs sphere conditioned medium was more effective in promoting the phenotypic transformation of macrophages from mainly pro-inflammatory M1 to anti-inflammatory M2, thereby playing an active anti-inflammatory role [195]. In 3D cultures, human pluripotent stem cells can be cultured to produce a trophoblast-like tissue model, which is helpful in analyzing the potential mechanisms of early human placental development [196]. Additionally, the 3D culture of stem cells includes self-assembly aggregation and encapsulation by hydrogel microspheres. Passanha et al. found that cells encapsulated by alginate hydrogel exhibited better viability on the 14th day, probably because the alginate gel could provide more room for cells and allow for the diffusion of more nutrients and oxygen [197].

The cytoskeleton of the internal structure of the cell provides the basis for the cell’s ability to sense external mechanical rigidity, thus mediating the interactions between the cell and the extracellular environment [198]. Some studies have even suggested that the hypoxic microenvironment may be the leading factor in the enhancement of the therapeutic potential of MSCs in 3D environments [199]. Chen et al. believed that a 3D environment’s conduciveness to the therapeutic effect of stem cells cannot be attributed only to the hypoxic microenvironment. Notably, the slight metabolic conversion of 3D cultured stem cells may have a significant impact on their paracrine potential. Upon further exploration, we found that 3D MSCs spheres made by 40,000 cells have the best paracrine and immune regulation ability [200]. Therefore, we recommend using microsphere material as a scaffold material to better complete tissue repair.

Organoids are micro-organs with three-dimensional structure grown in vitro using adult stem cells. Their genetic background and histological characteristics are very similar to those of in vivo organs; they have complex structures similar to those of real organs and can partially mimic the physiological functions of the source tissues and organs, which has made them a hot research topic in recent years [201]. A popular cutting-edge technology, stem cell populations propagated through an organoid can replace damaged or diseased tissues for the purpose of treating diseases. It is expected that when this technology reaches a certain stage of development, doctors will use pluripotent stem cells to build organoids that can replace damaged tissues in patients. Scientists can use human-derived stem cells to treat fractures after growing osteogenic-like organs in vitro. A segment of bone unit containing cortical bone, cancellous bone, and a junction interface, injected into the site of a bone defect, can spontaneously assemble and thus heal the fracture [202,203]. Low concentration gelatin methacrylate (GelMA) is a synthetic material that has excellent biocompatibility with cellular structures, and Xie et al. were able to rapidly print uniform GelMA microdroplets measuring approximately 100 μm using an electro-assisted bioprinting method. Due to the low external force applied to separate the droplets, the printing process results in minimal cell damage and can provide an excellent microenvironment for stem cell growth [204]. Alginate, chitosan, cell-laden PEG hydrogel, and other similar natural polymers are compatible with electrohydrodynamic spraying (EHS), and can all be used to generate controlled sized cell-loaded microspheres for potential applications in cell delivery and organoid culture [205]. PDMS microspheres with different elastic modulus (34 kPa, 0.6 MPa, and 2.2 MPa) were prepared and doped into MSCs, thus modulating the mechanical properties of the MSCs growth microenvironment and affecting MSCs differentiation, as MSCs differentiation is tension dependent [206]. It has been shown that MSCs in microspheres exposed to low contractile force are more conducive to adipose tissue differentiation than those exposed to high tension, which are more conducive to osteogenic differentiation [207]. When PDMS microspheres were incorporated into the cell sphere, the interfacial tension was increased due to the decrease in adhesion and the decrease in cadherin [208]. In addition, the introduction of material supports or spacers into the cell sphere increases the cell’s free volume, thus allowing a more frequent exchange of oxygen, nutrients, and waste with the exterior of the cell, thus facilitating cell growth and differentiation [209].

Preparation of cell-laden microspheres by 3D printing is an efficient means to achieve rapid construction of functional large-scale organoids [210]. Current 3D bioprinting technologies for manufacturing different types of organs mostly use droplet-based, extrusion-based, laser-induced forward transfer, and stereolithography bioprinting. Among them, extrusion-based 3D printing is the most popular choice. Hydrogel materials have become the obvious choice for 3D printed microspheres, which provide a highly hydrated environment with excellent biocompatibility. Bulk hydrogels have good shear-thinning properties for extrusion-based bioprinting, whereas granular hydrogels can only achieve this rheological property if they are “jammed” [211]. Jamming means that the particle system changes from a fluid to a solid state. When the particle-to-volume fraction reaches (Φ) to approximately 0.58, the jamming transition can be found. In theory, the monodispersed hard-spherical particles should be maximally jammed in a random configuration at Φ = 0.64 and be perfectly packed at Φ = 0.74. However, packaging of soft hydrogel particles is much more complicated due to inter-particle friction, charge, hardness, and microgel heterogeneity. In addition to this, microscale pores are necessary for cell growth. Therefore, there is currently an unmet need to develop granular jammed hydrogels that can preserve the microscale pores [212]. Ataie and co-workers programmed microgels for reversible interfacial nanoparticle self-assembly, enabling the fabrication of nanoengineered granular bioinks with well-preserved microporosity, enhanced printability, and shape fidelity [213]. Four-dimensional printing is a rapidly emerging field that has been developed from 3D printing, where printed structures change in shape, properties, or function when exposed to predetermined stimuli, such as humidity and temperature photoelectric stimuli. Four-dimensional bioprinting is a promising method to build cell-laden constructs that have complex geometries and functions for tissue/organ regeneration applications. A single-component jammed micro-flake hydrogel has been developed for 4D bioprinting with shear-thinning, shear-yielding, and rapid self-healing properties. As such, it can be printed smoothly as a stable three-dimensional biological structure when a photoinitiator and a UV absorber are added, and a gradient in cross-linking density is formed. After performing shape deformation, the hydrogel produced well-defined configurations and high cell viability, which may provide a number of applications in bone tissue engineering [214].

### 5.2. Endochondral Ossification for the Reparation of Large Bone Defects

Endochondral ossification has become an effective strategy in the repair of large segmental bone defects. The process of endochondral ossification occurs through the formation of a calcified cartilage matrix containing hypertrophic chondrocytes. MSCs undergo cartilage template formation, blood vessel formation, and mineralization. This avascular cartilage template secretes angiogenic factors that induce vascular growth in the defect site, a process that also activates osteogenic signaling pathways and ultimately promotes their restructuring into bone [145]. Mikael et al. combined gel phase with load-bearing and porous biodegradable matrices to produce hybrid matrices, with the PLGA polymer matrix providing mechanical stability, while the (hyaluronic acid-fibrin) gel-phase was expected to support the seeded MSC chondrogenesis, hypertrophy, and bone formation [215]. Endochondral ossification was achieved by incorporating gelatin particles capable of relatively rapid release of TGF-β1 and mineral-coated hydroxyapatite particles, allowing a more sustained release of BMP-2 into MSCs aggregates, in order to take advantage of the sequential release of dual growth factors. This method did not need long-term in vitro chondrogenic priming, and it exhibited greater mineralization than pure cell aggregates treated with exogenous growth factors [216]. Lin et al. applied the mechanism of endochondral ossification and co-cultured human umbilical vein endothelial cells with MSCs to create pre-vascularized bone-like tissue, which exhibited greater potential than either of the two cells cultured separately.

Xie et al. constructed a 3D culture system based on hydrogel microspheres using digital light-processing, which shows excellent in vitro stepwise-induction of BMSCs and achieves a state of simultaneous proliferation and differentiation of stem cells at the transcriptional level, as well as closely mimics the osteogenesis process. It also closely mimics the composition and behavior of stem progenitor cells in cartilage involved in osteogenesis. After in situ implantation, rapid bone repair was achieved within 4 weeks by advancing the regenerative process of endochondral osteogenesis, which indicates that in vitro construction of osteo-callus-like organs (osteo-callus organoids) according to developmental or regenerative processes is an effective strategy to promote rapid regeneration of bone tissue and the healing of bone defects [217]. Ji et al. developed a “building block” controlled drug delivery system with macrophage modulating and injectable properties consisting of porous chitosan (CS) microspheres and hydroxypropyl chitosan (HPCH) temperature-sensitive hydrogels, in which dimethoxyglycine (DMOG) was loaded into temperature-sensitive HPCH hydrogels and KGN was grafted onto porous CS microspheres, thus forming the HPCH hydrogels/CS microspheres composite scaffold, which can effectively regulate the regenerative microenvironment at the defect site, recruit autologous stem cells, and provide abundant cell adhesion sites, thus achieving M2 polarization of local macrophages and promoting osteochondral regeneration [218]. Janus particles are structures that integrate two or more composites with different chemical compositions into a single structural system. Janus particles can integrate different functional properties and perform multiple synergistic functions simultaneously because of their asymmetric composition and structure [80]. The application of Janus particles allows for the controlled release of drugs, and therefore it has promising applications in bone tissue engineering based on endochondral ossification.

### 5.3. Construction of Microspheres Integrating Multiple Functions

The generation of microspheres in bone tissue engineering is a fast-growing field, renowned for its therapeutic effects in regenerative medicine, particularly in the regeneration and replacement of bones. However, these microsphere structures are not limited to tissue regeneration, and have been found to have additional functions in recent years [219]. The “clean-to-repair” rhythm in the dynamic pathological osteoimmune microenvironment is essential to bone healing but is often disturbed by intense inflammation. Extracellular bioactive cations (Mg^2+^) were doped into (PLGA)/MgO-alendronate microspheres, and the microspheres effectively regulated the microenvironment by inducing the polarization of macrophages from the M0 to the M2 phenotype and enhancing the production of anti-inflammatory (IL-10) and osteopathic (BMP-2 and TGF-β1) cytokines. This has been proven to promote proliferation and osteogenic differentiation of bone marrow mesenchymal stem cells and produce a good bone regenerating effect [220,221]. Liang et al. combined microfluidic microspheres and self-assembled collagen nanofibers to form an injectable porous microsphere with a hierarchical micro/nanostructure that mimics the extracellular matrix and activates integrin-mediated macrophage (Mφ) polarization, resulting in microenvironmental reprogramming for paracrine transformation [222]. The microenvironment of bone defects resulting from some diseases produces large amounts of ROS, which not only impairs the regenerative potential of endemic stem cells but also reduces the therapeutic efficacy of stem cells implanted in the defect area. Excess ROS or insufficient antioxidants can make these bone defects difficult to repair. Therefore, for the treatment of refractory bone healing, there is an urgent need for new strategies to fabricate antioxidant microspheres that function as stem cell carriers to promote bone regeneration in the ROS microenvironment [223]. Yang et al. constructed fullerol-hydrogel microfluidic spheres by using a one-step innovative microfluidic technique for in situ regulation of redox homeostasis in stem cells and promotion of refractory bone healing [224]. Zheng et al. produced bone targeting antioxidative nano-iron oxide for treating postmenopausal osteoporosis [225]. Strategies now available for ROS scavenging to regulate the microenvironment include inorganic nanoparticles (cerium oxide nanoparticles, iron oxide nanoparticles, manganese nanoparticles, and carbon nanomaterials) and organic moieties (phenol group, sulfur, and boronic acid) [226].

The incorporation of antibiotics and some active ingredients from plants, such as cinnamaldehyde and nanosilver, into microspheres gives the microspheres antibacterial effects, thus making them a viable treatment for osteomyelitis bone defects [227,228,229,230]. Meanwhile, the incorporation of the antitumor drug adriamycin into microspheres for bone tissue engineering can achieve long-term release, resulting in bone regeneration and the destruction of tumor cells, and thus shows good prospects for application [231]. The doping of M-type ferrite particles (SrFe_12_O_19_) into the microspheres enables the rapid release of doxorubicin (DOX) under the irradiation of near-infrared laser light, enabling the promotion of osteogenesis and enhancing synergistic photothermal chemotherapy against osteosarcoma [232]. Bleeding due to bone defects is a clinically recognized category of refractory bleeding, wherein damage to the abundant blood vessels in cancellous bone leads to continuous bleeding (oozing) during the slow bone repair process. Therefore, Liu et al. devised a method to construct multilayer structured microspheres based on branched-chain starch, porous starch, and tannic acid of biomass (maize and grape) origin and MQ_2_T_2_ with the outmost polyphenol layer having unique platelet adhesion, activation, and erythrocyte aggregation properties. The microspheres had excellent hemostatic and pro-bone repair properties with good therapeutic effects in both rat and beagle models of cancellous bone defects. They also had excellent hemostatic and pro-bone repair properties [233].

## 6. Discussion

In this review, we comprehensively describe the microsphere preparation methods, classification of raw materials, and drug loading techniques and analyze several new strategies for using microspheres for bone tissue engineering. We believe this new "bottom-up" modular tissue engineering technique offers advantages over conventional monolithic scaffolds regarding improved continuous delivery of growth factors, and the manipulation of internal cells. Advances are expected in all areas discussed, especially new manufacturing strategies to prepare more abundant microspheres, bringing further enthusiasm for their application.

Regarding the fabrication of microspheres, technology will continue to improve to scale up the fabrication of microspheres, prepare homogeneous particles, and achieve batch-to-batch consistency at high productivity [234]. In recent years, these goals have been progressively achieved using microfluidic platforms by controlling precursor flow and mixing, and Janus particles with heterogeneity have been prepared on microfluidic platforms. With these advances and improved material consistency, the choice of materials for preparation has been diversified, and stimuli-responsive and phase-change materials have been introduced to achieve greater functionality under endogenous and exogenous stimuli [80]. Computational models and numerical methods have also been developed to predict the properties and behavior of microspheres, as well as the local internal and characteristic structures [235].

Microsphere structures have great potential for scalable seed cells for bone tissue engineering, and the ability to carry drugs and various biologics makes multifunctional microspheres central to the future development of bone tissue regeneration. In addition, the use of these methods allows for the encapsulation of many different therapeutic agents in microspheres, resulting in a more direct induction of surrounding seed cells. The use of controlled release or sustained release can also work locally, more in line with normal physiological processes, thus providing a suitable microenvironment. Furthermore, with the rapid development of stem cell and drug design technologies, the options for microsphere cargo will continue to increase, such as combining genetic engineering and tissue engineering [236]. With the increasing understanding of bone reconstruction, repair of bone defects with nucleic acid-based gene therapy is emerging as a promising strategy for restoring anatomical structure, appearance, and function to the injured site. Microsphere structures have great potential to deliver sequential signals to tissues and cells, including exosomes and encoding DNA (including pDNA), mRNA, miRNA, small interfering RNA (siRNA), and small hairpin RNA (shRNA). The promotion of systematization, standardization, non-toxicity, harmlessness, and better biocompatibility of artificial microspheres, the expanding organic integration of microsphere products with traditional therapeutic approaches, and emerging stem cell technologies are creating a substantial medical and scientific market.

Microspheres also have the potential to be developed with new and interesting properties, such as the 3D culture of seed cells and construction of organoids, and more importantly, the preparation of cell-laden microspheres by 3D printing is a powerful means to achieve rapid construction of large-scale functionalized organoids. Interference transitions in granular hydrogels are widely used to achieve a potential paradigm shift in extrusion-based 3D bioprinting. Shear thinning and self-healing provide better physiological conditions for 3D printing and organoid construction. In the construction of bone tissue engineering, critical-size bone defects often face limited self-healing ability and long recovery times. The concept of “developmental engineering” has been proposed to mimic the critical developmental events in endochondral osteogenesis to promote more efficient bone tissue regeneration. Microsphere-based 3D culture systems can highly mimic the composition and behavior of stem progenitor cells during osteogenesis in cartilage, leading to efficient bone repair. Unlike conventional tissue engineering that focuses only on tissue repair, microsphere systems integrate additional functions, such as regulation of the immune system, anti-inflammation, and hemostasis, in a single microsphere to synergize and promote tissue repair have been constructed.

Of course, despite these advantages, there are also many problems with practical ap-plications that we need to address. Firstly, although the development of preparation methods and materials has made the manufacture of microspheres easier and more diverse, the harsh storage methods and relatively expensive prices are still problems that need to be addressed in clinical applications. Secondly, although some of the materials and preparation equipment used by scientists are suitable for the human body and have good biocompatibility, some of the raw materials used are cytotoxic, and the degradation products may not be suitable for microenvironment stabilization. The preparation process is also toxic and pathogenic. All these factors limit the application of microspheres in clinical, cell biology, and regenerative medicine. In addition, due to the plasticity and low viscosity of the microsphere material, microspheres can be used in the treatment of various types of complex defects. However, their shape and mechanical properties are poor, making them unsuitable for areas with high mechanical bearing, and they often have to be used together with stent materials. Finally, with the rapid development of stem cell regenerative medicine, regulatory issues such as social ethics and laws governing the preparation and application of various materials also deserve our attention.

## Figures and Tables

**Figure 1 pharmaceutics-15-00321-f001:**
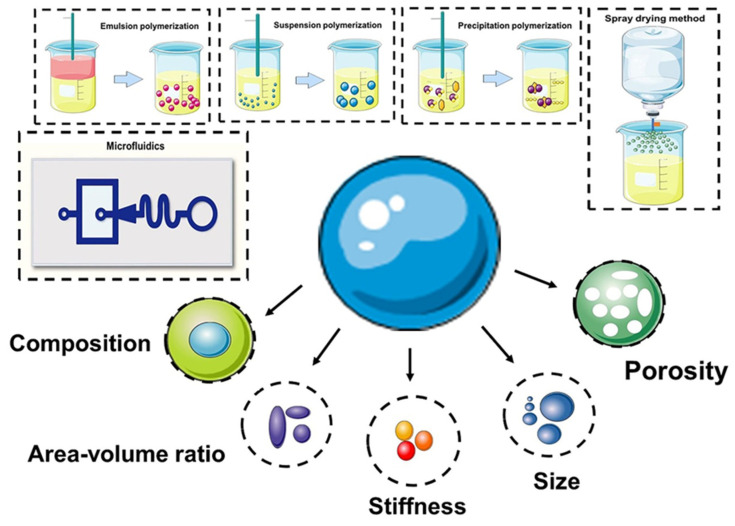
Preparation of microspheres and (bio)chemical/physical features as modulating moieties of microspheres.

**Figure 2 pharmaceutics-15-00321-f002:**
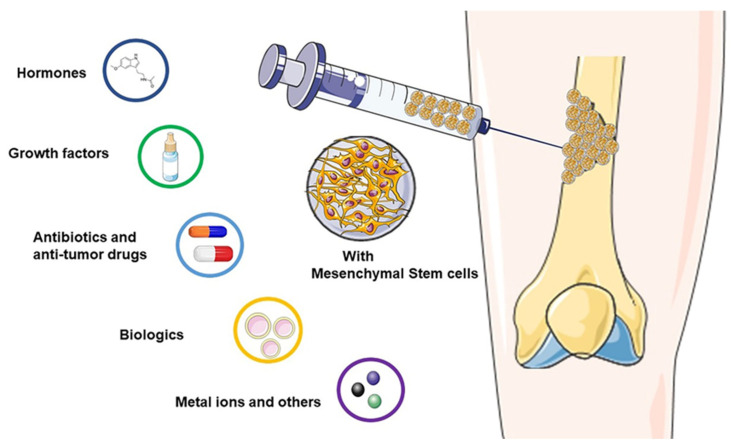
Microspheres as carriers in bone tissue engineering.

**Table 1 pharmaceutics-15-00321-t001:** Products loaded by microspheres for bone tissue engineering.

Type	Name	Reference
Growth factors	bone morphogenic protein-2 (BMP-2)	[83]
	bone morphogenetic factor-6 (BMP-6)	[84]
	interleukin-4 (IL-4)	[85]
	vascular endothelial growth factor (VEGF)	[59]
	insulin-like growth factor (IGF-1)	[86]
	fibroblast growth factor (FGF)	[87]
	fibroblast growth factor-2 (FGF-2)	[88]
	osteogenic growth peptide (OGP)	[89]
	stromal cell derived factor-1 (SDF-1)	[90]
	platelet-derived growth factor (PDGF)	[91]
	transforming growth factor-β1 (TGF-β1)	[92]
Antibiotics	gentamicin sulfate (GS)	[93]
	ciprofloxacin	[94]
	Vancomycin	[95]
	levofloxacin	[96]
	gentamicin	[97]
	tetracycline	[98]
	metronidazole	[99]
	doxycycline	[100]
	colistin	[101]
Antineoplastic drugs	methotrexate (MTX)	[102]
	doxorubicin (DOX)	[103]
	paclitaxel (PTX)	[104]
	etoposide (ETP)	[105]
Anti-tuberculosis drugs	isoniazid	[106]
	rifampin	[107]
Hormone	dexamethasone	[108]
	melatonin	[109]
Biologics	non-coding microRNA	[110]
	exosomes	[111]
Metals	trontium ranelate (SrR)	[112]
	silver nanoparticles (Ag NPs)	[113]
	Co ions	[114]
	Strontium (Sr)	[115]
Others	alendronate (ALN)	[116]
	salvianolic acid B (SB)	[117]
	Kartogenin (KGN)	[118]
	simvastatin	[119]
	oxygen carrier perfluorooctane (PFO)	[120]
	parthenolide	[121]
	naringin	[122]
	lovastatin	[123]
	nano-sized hydroxyapatite (HAP)	[124]
	ketoprofen	[125]
	heparin	[126]
	kaempferol (KEM)	[127]

## Data Availability

Data is contained within the article.

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
