# Peer review of "The Role of Microsphere Structures in Bottom-Up Bone Tissue Engineering"

_pharmaceutics, 2023, doi:10.3390/pharmaceutics15020321_

Round 1

Reviewer 1 Report

In this review article, Feng et al. provided an overview of biomaterial microspheres for the use of bone tissue engineering. Microsphere fabrication methods and the use of microspheres as drug carriers, bioreactors, and reservoirs are summarized and discussed. Some newly developed strategies based on microspheres are also reviewed.

This review article is written in a logical layout that is easy to read and understand and provides a clear clue for the microsphere materials and their use in bone tissue engineering, albeit certain improvements for each subsection can be made. For example, in the materials part, materials (e.g., structures and components) and their physicochemical properties (toxicity, degradability, and mechanical properties) should be briefly introduced. In addition to the mentioned natural materials and synthetic materials such as chitosan and PLGA, other representative polysaccharide and protein-based natural materials such as alginate and its derivatives and gelatin and its derivatives and synthetic materials such as PEG derivatives should be also succinctly described.

In addition to the physicochemical properties, an important consideration of microspheres as scaffolding materials is their unique mechanical properties. Some important review articles can be a good reference for this part, e.g.,  https://doi.org/10.1038/s41578-019-0148-6. This can also be briefly described somewhere in the manuscript, without changing the layout of the current manuscript.

Specific to the new application section, a trending topic regarding the use of hydrogel microspheres as biomaterials for bone tissue engineering is the emerging 3D/4D bioprinting (https://doi.org/10.1002/VIW.20200060https://doi.org/10.1002/adma.202109394https://doi.org/10.1002/smll.202202390).

To enable further manipulation of mechanical properties and porosity, hydrogel microspheres are further processed into a jammed state (https://doi.org/10.1016/j.copbio.2018.11.001). This special microsphere could also be mentioned in a few sentences either in the introduction part or the perspective part.

Generally, this review article is well organized and could be a good reference for readers who are interested in hydrogel microsphere materials. This article is thus recommended for publication after addressing the minor concerns mentioned above.  

Author Response

Response to Reviewer 1 Comments

In this review article, Feng et al. provided an overview of biomaterial microspheres for the use of bone tissue engineering. Microsphere fabrication methods and the use of microspheres as drug carriers, bioreactors, and reservoirs are summarized and discussed. Some newly developed strategies based on microspheres are also reviewed.

Point 1: This review article is written in a logical layout that is easy to read and understand and provides a clear clue for the microsphere materials and their use in bone tissue engineering, albeit certain improvements for each subsection can be made. For example, in the materials part, materials (e.g., structures and components) and their physicochemical properties (toxicity, degradability, and mechanical properties) should be briefly introduced. In addition to the mentioned natural materials and synthetic materials such as chitosan and PLGA, other representative polysaccharide and protein-based natural materials such as alginate and its derivatives and gelatin and its derivatives and synthetic materials such as PEG derivatives should be also succinctly described.  

Response 1: Thank you for your comments, we have revised the manuscript according to your suggestion. The revised manuscript is quoted as follows:

“Chitosan can have its physical properties modified by various chemical modifications, such as carboxymethylation, thiolation, and succinylation. Currently, synthesized chitosan derivatives have improved solubility, increased cellular uptake, reduced cytotoxicity, encapsulated different types of drugs, and sustained release properties [49]. In addition, chitosan of different molecular weights has different biological efficacy. For example, a study showed that microspheres of high molecular weight chitosan exhibited lower adhesion and lower MTX release compared to medium molecular weight chitosan rates [50].”

“A study by Dennis P Link et al. showed that gelatin type affects the degradation of gelatin particles incorporated into calcium phosphate (CaP) cement. However, this difference in degradation and the consequent macroporosity did not induce a difference in biological response [54]. Gelatin and chondroitin sulfate microspheres can be degraded by several gelatinases found in the synovial fluid of osteoarthritic joints [55]. In addition, enzymatic degradation of gelatin/HA microspheres may lead to reduced calcium uptake by reducing the integrity of the composite microspheres [56].

Alginate-based materials have received considerable attention in biomedical applications due to their hydrophilicity, biocompatibility, and physical structure. Applications include cell encapsulation, drug delivery, stem cell culture, and tissue engineering scaffolds [57]. Alginates have been analyzed for tissue engineering applications along with bone morphogenetic proteins, vascular endothelial growth factor, transforming growth factor β-3, other growth factors, cells, proteins, drugs, and osteoinductive agents [58]. For example, several authors have developed injectable hybrid RGD-alginate/Laponite (RGD-Alg/Lap) hydrogel microspheres to co-encapsulate human dental pulp stem cells (hDPSCs) and vascular endothelial growth factor (VEGF). The microspheres exhibited tunability in terms of mechanical properties and sustained-release capability capacity. The incorporation of hectorite and sustained release of VEGF supported not only the differentiation of dental pulp stem cells in vitro but also the regeneration of new tissues in vivo [59].

Chitosan, collagen, and starch are the matrix components of the tissue, all of which have good biocompatibility and biodegradability, low cytotoxicity, and degradation products that can be completely absorbed by the body. However, the disadvantage with these components is that they have poor mechanical strength, and their degradation rates are not consistent. Sophie R. Van Tommeden et al. reported the degradation behavior of in situ gelling hydrogel matrices consisting of positively and negatively charged dextran microspheres. They found that the degradation of positively charged microspheres was faster compared to negatively charged microspheres due to the presence of protonated tertiary amine groups in the cationic microspheres, which stabilize the transition state during hydrolysis. On the other hand, the presence of negatively charged groups leads to the repulsion of hydroxyl anions, which results in a slower degradation rate [60].”

“Other synthetic microsphere materials such as poly(ethylene glycol) (PEG) or its derivatives and the combinations of multiple polymers are also favored by scientists. For example, some scientists have used a combination of chitosan and polyethylene glycol diacrylate (PEGDA) to use chondrocyte-loaded microspheres as a cell carrier based on a double network hydrogel. Applying microfluidic technology, size-controlled chitosan/PEGDA hydrogel microspheres (CP-MSs) were fabricated by the water-in-oil method after photo-crosslinking and physical cross-linking. Chondrocytes loaded on CP-MSs showed good cell viability and proliferation after long-term cell culture [64]. Microspheres with surface micropores were obtained by using an amphiphilic triblock copolymer (PLLA-PEG-PLLA) consisting of poly(L-lactic acid) (PLLA) and poly(ethylene glycol) (PEG) segments. The W1/O/W2 double emulsification method was used. When the PEG fraction was controlled at 10 wt.%, the microspheres exhibited higher cell affinity than the smooth-surfaced PLLA microspheres. After further functionalization with polydopamine coating and apatite deposition, PLLA-PEG-PLLA microspheres significantly enhanced the osteogenic differentiation of bone marrow mesenchymal stem cells (BMSCs) [65].”

And the additions and revisions were highlighted on line 257 to 328 in the revised manuscript.

Point 2: In addition to the physicochemical properties, an important consideration of microspheres as scaffolding materials is their unique mechanical properties. Some important review articles can be a good reference for this part, e.g.,  https://doi.org/10.1038/s41578-019-0148-6. This can also be briefly described somewhere in the manuscript, without changing the layout of the current manuscript.

Response 2: Thanks to your suggestion, we have reorganized the introduction to include the unique mechanical properties of microspheres. The revised manuscript is quoted as follows:

“Microspheres have a number of unique properties compared to large pieces of material materials that make them attractive for biomedical applications. Microspheres can be fabricated from both natural and synthetic materials, and can be made to vary in shape, density, porosity, and size by applying techniques that are often compatible with the encapsulation of biologics. Microspheres for different application scenarios require different properties [24]. Microspheres can be divided into three categories: suspensions, granular and composites. In suspensions, the microspheres reside in a fluid (liquid or air), with minimal interactions between particles. When the particle-packing density increases, granular microspheres form. If microspheres are embedded within a bulky material, a composite is obtained [25]. Physical interactions between microspheres typically result in shear thinning behavior and solid consistency without chemical modification after injection. Not only that, microspheres are inherently modular, because multiple microspheres populations can be mixed together to create different materials with different properties [26]. “

And the additions and revisions were highlighted on line 81 to 94 in the revised manuscript.

Also, a discussion of this section has been added to the DISCUSSION section.

Point 3: Specific to the new application section, a trending topic regarding the use of hydrogel microspheres as biomaterials for bone tissue engineering is the emerging 3D/4D bioprinting (https://doi.org/10.1002/VIW.20200060https://doi.org/10.1002/adma.202109394https://doi.org/10.1002/smll.202202390).

Response 3: Thank you for the suggestion which is of great significance to improve the manuscript. We have added “3D/4D bioprinting” part on line 642 to 672 in the revised manuscript which were highlighted.

The revised manuscript is quoted as follows:

“Preparation of cell-laden microspheres by 3D printing is an efficient means to achieve rapid construction of functional large-scale organoids [210]. Current 3D bioprinting technologies for manufacturing different types of organs mostly use droplet-based, extrusion-based, laser-induced forward transfer, and stereolithography bioprinting. Among them, extrusion-based 3D printing is the most popular choice. Hydrogel materials have become the obvious choice for 3D printed microspheres, which provide a highly hydrated environment with excellent biocompatibility. Bulk hydrogels have good shear-thinning properties for extrusion-based bioprinting, while granular hydrogels can only achieve this rheological property if they are "jammed"[211]. Jamming means that the particle system changes from a fluid to a solid state. When the particle-to-volume fraction reaches (Φ) to approximately 0.58, the jamming transition can be found. In theory, the monodispersed hard-spherical particles should be maximally jammed in a random configuration at Φ = 0.64 and be perfectly packed at Φ = 0.74. However, packaging of soft hydrogel particles is much more complicated due to inter-particle friction, charge, hardness and microgel heterogeneity. In addition to this, microscale pores are necessary for cell growth. Therefore, there is currently an unmet need to develop granular jammed hydrogels that can preserve the microscale pores [212]. Ataie and co-workers programmed microgels for reversible interfacial nanoparticle self-assembly, enabling the fabrication of nanoengineered granular bioinks (NGB) with well-preserved microporosity, enhanced printability, and shape fidelity [213]. 4D printing is a rapidly emerging field that has been developed from 3D printing, where printed structures change in shape, properties or function when exposed to predetermined stimuli, such as humidity and temperature photoelectric stimuli. 4D bioprinting is a promising method to build cell-laden constructs that have complex geometries and functions for tissue/organ regeneration applications. A single-component jammed micro-flake hydrogel has been developed for 4D bioprinting with shear-thinning, shear-yielding, and rapid self-healing properties. As such, it can be printed smoothly as a stable three-dimensional biological structure when a photoinitiator and a UV absorber are added, and a gradient in cross-linking density is formed. After performing shape deformation, the hydrogel produced well-defined configurations and high cell viability, which may provide a number of applications in bone tissue engineering [214]. ”

Point 4: To enable further manipulation of mechanical properties and porosity, hydrogel microspheres are further processed into a jammed state (https://doi.org/10.1016/j.copbio.2018.11.001). This special microsphere could also be mentioned in a few sentences either in the introduction part or the perspective part.

Response 4: Thank you for this suggestion. The hydrogel microspheres which are further processed into a jammed state has been described on line 642 to 672 in the revised manuscript. And the detailed revisions have been stated in Q3.

Generally, this review article is well organized and could be a good reference for readers who are interested in hydrogel microsphere materials. This article is thus recommended for publication after addressing the minor concerns mentioned above.  

We are most thankful for the reviewers' positive and constructive evaluations rendered to our manuscript. We have adhered to their advices, and we are convinced that their assistance has helped to improve the quality of the manuscript.

Thank you for handling our manuscript.

With best regards,

Shu Guo

Department of Plastic Surgery

The First Hospital of China Medical University, Shenyang 110002, China

Reviewer 2 Report

The work is interesting, but along with the analysis of a huge number of references (220 sources), the authors made unfortunate mistakes:

- The design of the work is difficult to understand, since the authors were unable to build an integral consistent and logical system of presentation of the material, arranging the text using patches of analyzed literature references.

- As a result of poor design, the work looks fragmented. During the presentation of the text, the authors often break off one thought and abruptly move on to another, sometimes even within the same paragraph. Not infrequently such a transition is devoid of logical meaning.

- The work lacks a qualitative introductory section containing basic information on the topic of microspheres, including definitions, classifications, history of the issue, as well as the development of technology.

- The "Discussions" section is extremely small and contains loud slogans, not an analysis of the material. It could be renamed "Conclusion", but since this is a review article, the authors still need to leave exactly the "Discussion", significantly supplement, and rework it.

- The English language of the manuscript is weak, looks like a machine translation from Chinese with adaptation, as evidenced by the remnants of hieroglyphs found in the text.

- Most of all, I was upset by the huge number of abbreviations without decoding and without the “abbreviations” section at the beginning or end of the manuscript, which is a manifestation of disrespect for readers.

Author Response

Response to Reviewer 2 Comments

The work is interesting, but along with the analysis of a huge number of references (220 sources), the authors made unfortunate mistakes:

Point 1: The design of the work is difficult to understand, since the authors were unable to build an integral consistent and logical system of presentation of the material, arranging the text using patches of analyzed literature references.

Response 1: Bottom-up tissue engineering is an effective strategy for bone defect repair. Bone tissue engineering is an interdisciplinary biomedical engineering method that involves scaffold materials, seed cells, and "growth factors. Microspheres are promising for biomedical applications, ranging from the therapeutic delivery of cells and drugs to the production of scaffolds for tissue repair and bioinks for 3D printing. We have covered the main classes of microspheres systems, the fabrication techniques to manufacture them, their properties across length scales and a number of their biomedical applications. In addition, some new strategies in bone tissue engineering construction have been mentioned in this review. When readers want to make a microsphere structure for the construction of bone tissue engineering, they can refer to this review.

Point 2: As a result of poor design, the work looks fragmented. During the presentation of the text, the authors often break off one thought and abruptly move on to another, sometimes even within the same paragraph. Not infrequently such a transition is devoid of logical meaning.

Response 2: We apologize for the writing issues in our manuscript and thanks for your valuable suggestions, we have reorganized the introduction and discussion to increase the coherence of the article. For the main part of the article, we have made appropriate deletions and corrections to increase the integrity and continuity of the article. We will be happy to edit the text further, based on helpful comments from the reviewer.

Point 3: The work lacks a qualitative introductory section containing basic information on the topic of microspheres, including definitions, classifications, history of the issue, as well as the development of technology.

Response 3: Thanks to your suggestion, we have reorganized the introduction to include the basic information on the topic of microspheres. The revised manuscript is quoted as follows:

“Microspheres have a number of unique properties compared to large pieces of material materials that make them attractive for biomedical applications. Microspheres can be fabricated from both natural and synthetic materials, and can be made to vary in shape, density, porosity, and size by applying techniques that are often compatible with the encapsulation of biologics. Microspheres for different application scenarios require different properties [24]. Microspheres can be divided into three categories: suspensions, granular and composites. In suspensions, the microspheres reside in a fluid (liquid or air), with minimal interactions between particles. When the particle-packing density increases, granular microspheres form. If microspheres are embedded within a bulky material, a composite is obtained [25]. Physical interactions between microspheres typically result in shear thinning behavior and solid consistency without chemical modification after injection. Not only that, microspheres are inherently modular, because multiple microspheres populations can be mixed together to create different materials with different properties [26]. “

And the additions and revisions were highlighted on line 81 to 94 in the revised manuscript.

Point 4: The "Discussions" section is extremely small and contains loud slogans, not an analysis of the material. It could be renamed "Conclusion", but since this is a review article, the authors still need to leave exactly the "Discussion", significantly supplement, and rework it.

Response 4: Thank you for the suggestion which is of great significance to improve the manuscript. We have rewritten the discussion section part on line 776 to 821 in the revised manuscript.

The revised manuscript is quoted as follows:

“In this review, we comprehensively describe the microsphere preparation methods, classification of raw materials, and drug loading techniques and analyze several new strategies for using microspheres for bone tissue engineering. We believe this new "bottom-up" modular tissue engineering technique offers advantages over conventional monolithic scaffolds regarding improved continuous delivery of growth factors, and the manipulation of internal cells. Advances are expected in all areas discussed, especially new manufacturing strategies to prepare more abundant microspheres, bringing further enthusiasm for their application.

Regarding the fabrication of microspheres, technology will continue to improve to scale up the fabrication of microspheres, prepare homogeneous particles, and achieve batch-to-batch consistency at high productivity [234]. In recent years, these goals have been progressively achieved using microfluidic platforms by controlling precursor flow and mixing, and Janus particles with heterogeneity have been prepared on microfluidic platforms. With these advances and improved material consistency, the choice of materials for preparation has been diversified, and stimuli-responsive and phase-change materials have been introduced to achieve greater functionality under endogenous and exogenous stimuli [80]. Computational models and numerical methods have also been developed to predict the properties and behavior of microspheres, as well as the local internal and characteristic structures [235].

Microsphere structures have great potential for scalable seed cells for bone tissue engineering, and the ability to carry drugs and various biologics makes multifunctional microspheres central to the future development of bone tissue regeneration. In addition, the use of these methods allows for the encapsulation of many different therapeutic agents in microspheres, resulting in a more direct induction of surrounding seed cells. The use of controlled release or sustained release can also work locally, more in line with normal physiological processes, thus providing a suitable microenvironment. Furthermore, with the rapid development of stem cell and drug design technologies, the options for microsphere cargo will continue to increase, such as combining genetic engineering and tissue engineering [236]. With the increasing understanding of bone reconstruction, repair of bone defects with nucleic acid-based gene therapy is emerging as a promising strategy for restoring anatomical structure, appearance, and function to the injured site. Microsphere structures have great potential to deliver sequential signals to tissues and cells, including exosomes and encoding DNA (including plasmid DNA, pDNA), mRNA, miRNA, small interfering RNA (siRNA), and small hairpin RNA (shRNA). The promotion of systematization, standardization, non-toxicity, harmlessness, and better biocompatibility of artificial microspheres, the expanding organic integration of microsphere products with traditional therapeutic approaches, and emerging stem cell technologies are creating a substantial medical and scientific market.

Microspheres also have the potential to be developed with new and interesting properties, such as the 3D culture of seed cells and construction of organoids, and more importantly, the preparation of cell-laden microspheres by 3D printing is a powerful means to achieve rapid construction of large-scale functionalized organoids. Interference transitions in granular hydrogels are widely used to achieve a potential paradigm shift in extrusion-based 3D bioprinting. Shear thinning and self-healing provide better physiological conditions for 3D printing and organoid construction. In the construction of bone tissue engineering, critical-size bone defects often face limited self-healing ability and long recovery times. The concept of "developmental engineering" has been proposed to mimic the critical developmental events in endochondral osteogenesis to promote more efficient bone tissue regeneration. Microsphere-based 3D culture systems can highly mimic the composition and behavior of stem progenitor cells during osteogenesis in cartilage, leading to efficient bone repair. Unlike conventional tissue engineering that focuses only on tissue repair, microsphere systems integrate additional functions, such as regulation of the immune system, anti-inflammation, and hemostasis, in a single microsphere to synergize and promote tissue repair have been constructed.

Of course, despite these advantages, there are also many problems with practical ap-plications that we need to address. Firstly, although the development of preparation methods and materials has made the manufacture of microspheres easier and more diverse, the harsh storage methods and relatively expensive prices are still problems that need to be addressed in clinical applications. Secondly, although some of the materials and preparation equipment used by scientists are suitable for the human body and have good biocompatibility, some of the raw materials used are cytotoxic, and the degradation products may not be suitable for microenvironment stabilization. The preparation process is also toxic and pathogenic. All these factors limit the application of microspheres in clinical, cell biology, and regenerative medicine. In addition, due to the plasticity and low viscosity of the microsphere material, microspheres can be used in the treatment of various types of complex defects. However, their shape and mechanical properties are poor, making them unsuitable for areas with high mechanical bearing, and they often have to be used together with stent materials. Finally, with the rapid development of stem cell regenerative medicine, regulatory issues such as social ethics and laws governing the preparation and application of various materials also deserve our attention.”

Point 5: The English language of the manuscript is weak, looks like a machine translation from Chinese with adaptation, as evidenced by the remnants of hieroglyphs found in the text.

Response 5: We apologize for the writing issues in our manuscript. We have worked on both language and readability. We really hope that the language has been substantially improved in the newly submitted version. But the comment“hieroglyphics,” is confusing in so far as we see zero Chinese characters in this document or the older version.

Point 6: Most of all, I was upset by the huge number of abbreviations without decoding and without the “abbreviations” section at the beginning or end of the manuscript, which is a manifestation of disrespect for readers.

Response 6: We apologize for forgetting to include an abbreviation in the manuscript, we have absolutely no intention of insulting readers. The list of abbreviations has been added in line 839.

We are most thankful for the reviewers' positive and constructive evaluations rendered to our manuscript. We have adhered to their advices, and we are convinced that their assistance has helped to improve the quality of the manuscript.

We hope that the revised manuscript will fulfill the requirements for publication in Pharmaceutics.

Thank you for handling our manuscript.

With best regards,

Shu Guo

Department of Plastic Surgery

The First Hospital of China Medical University, Shenyang 110002, China

Round 2

Reviewer 2 Report

Dear Colleagues!

Thank you for the serious work, that have been done and for the corrections made to the initial text of your manuscript. I can say that the corrections take into account all the comments and suggestions marked by me after the first review. I completely satisfied with the comments of the authors. I want to ask the authors to be more diligent in preparing new manuscripts, wish them good luck and success in the future, and congratulate them on coming Chinese New Year!

I apologize for the slight delay in reviewing. This happened because of the long New Year's holidays.

PS. Dear authors, look at line number 436 in the initial manuscript and you will see there a Chinese hieroglyph 和 (and). It, by the way, remained in the corrected version. Pay attention to line 483. Correct it, please.

Best wishes!